# Practicability of a Time-Limited Welfare Assessment Protocol for Pasture-Based Dairy Farms, and a Preliminary Assessment of Welfare Outcome Thresholds

**DOI:** 10.3390/ani12182481

**Published:** 2022-09-19

**Authors:** Sujan Sapkota, Richard Laven, Kristina Ruth Müller, Nikki Kells

**Affiliations:** School of Veterinary Science, Massey University, Tennent Drive, Palmerston North 4474, New Zealand

**Keywords:** animal-based measures, welfare threshold development, locomotion scoring, paddock observation, milking cow

## Abstract

**Simple Summary:**

This study tested the feasibility and practicability of 32 welfare measures as a part of a time-limited protocol across 23 pasture-based farms. The study identified that one measure, maximum time waiting to enter the milking parlour, was required to be assessed alongside locomotion scoring (by an observer outside the milking parlour) and that our measures of water availability (distance between water points), and behaviour (30 min observation in paddock prior to milking) needed significant changes before inclusion in the protocol. Overall, the protocol was generally feasible, but further testing for repeatability and reliability of measures with multiple observers is needed.

**Abstract:**

This study assessed a new time-limited protocol developed for pasture-based cows across 23 dairy farms. The process started prior to milking with a questionnaire, followed by an assessment of resources (16 farms only) and behavioural observation of cows at pasture. Remaining animal-based measures were assessed during milking, usually by two assessors (one parlour based and one outside). The protocol proved to be practical and feasible with limited changes needed, except for the assessment of water availability and behaviour. As most cows could access only one water trough, distance between troughs was not a measure of water availability, while the observation of a large numbers of cows at pasture for 30 min resulted in few observations and an uncertain denominator (effective number of observed cows). Further research is needed to determine the best way of assessing water availability and cow behaviour in a time-limited assessment of pasture-based cows. Three animal-based measures (broken tails, dirtiness, and coughing) had mean values higher than the author-determined acceptable thresholds, while <50% of farms met trough cleanliness and track condition targets, and none met the criteria for shelter and shade. This was a sample of farms based on convenience, so more data are required to establish the representativeness of these results. Such testing should involve assessment of the repeatability and reliability of the measures in our protocol.

## 1. Introduction

Changes and advancements in animal production have resulted in public concern regarding the treatment of animals on farm and their quality of life. In response to this, animal-focused welfare-assessment protocols have become increasingly common, especially in Europe [1,2,3]

All assessment protocols must address the management used on the farm and use measures selected after appropriate feasibility testing [4]. This approach is particularly important in systems such as the pasture-based dairy farming system which predominates in New Zealand, which are very different from the housed systems which are the principal focus of most welfare assessment schemes. This is because many parameters useful in housed systems are not relevant or useful in cows that are never housed [5]. As there is currently no industry-standard welfare assessment scheme for dairy farms in New Zealand, we developed a welfare assessment protocol for use on pasture-based dairy farms in New Zealand [6]. This protocol was focused on being time-limited with measurements made during and around milking. It consisted of 32 on-farm measures, of which nine were resource-based, 13 were animal-based, six were record-based and two each were management- and stock person-based. The protocol required two people and took 2–3.5 h to complete (depending on the farm size). However, the feasibility and applicability of those measures needed further testing before the protocol could be recommended for use as a welfare assessment tool on New Zealand dairy farms.

One key challenge during the development of a protocol is the setting of thresholds for the measures used in the protocol. These thresholds can be used for a variety of purposes, from identifying farms not meeting compliance standards to identifying areas for action on farms that need to improve beyond the baseline, to identifying farms achieving high standards. However, for most measures, especially in understudied systems such as pasture-based dairy systems, there are no clearly defined thresholds indicative of acceptable and unacceptable welfare [7]. Indeed, it seems unlikely that universally agreed thresholds are feasible, as there are significant disagreements even between experts on what constitutes acceptable and unacceptable welfare [1,5]. Thus, any expert-set threshold will be a compromise, even if experts are carefully chosen [6]. However, there is increasing recognition that optimising animal welfare is a crucial part of farming’s “social licence” and that this social licence is determined by people other than experts [8]. Consideration of stakeholder opinions, including all purchasers of dairy products such as dairy companies, retailers, and consumers is thus important when setting welfare thresholds as much of the assessment is targeted at the social licence to farm. However, including consumers in the process is likely to significantly increase the variability in what is considered to be acceptable and the potential identification of different areas of concern [9,10,11] making the process of deciding thresholds and identifying areas of concern more complex and time consuming.

The alternative to using thresholds based on opinion (expert or otherwise) is the use of benchmarking, where an individual farm’s results for each indicator or measure can be compared to results from a larger group of farms (which may be national, local, or farm-type based). However, this seemingly simple approach requires a relatively large dataset, so that the benchmarking is robust [12]. Additionally, although benchmarking can drive industry change [13], it may fail to identify systemic problems, especially if thresholds are set using industry averages (e.g., broken tails on New Zealand dairy farms [14]).

Thus as a starting point, we created thresholds based on the authors’ opinions as to what was acceptable and unacceptable welfare. This is the same approach used to determine thresholds for a protocol for assessing beef cow welfare in New Zealand [15].

The aims of this study were (i) to test our previously developed time-limited protocol [6] across more New Zealand dairy farms, and (ii) to assess the data collected on the included measures against author-determined thresholds for a provisional assessment of animal welfare status on these farms.

## 2. Materials and Methods

### 2.1. Methods and Categorisation of Welfare Measures

The measurement methods for each of the 32 measures are summarised in Table 1, Table 2, Table 3 and Table 4, along with their threshold values. Due to the lack of established thresholds for welfare measures for pasture-based dairy cows [16], thresholds were set based on the authors’ opinion, guided, when available, by published recommendations. We assigned three categories of welfare to most measures: acceptable welfare (represented by green), marginal welfare (represented by orange), and unacceptable welfare (represented by red) as in Kaurivi et al. [15], but where marginal welfare was thought to be an inappropriate category, e.g., for the question “do you use pain relief for disbudding”, we assigned only two categories (red and green) (see Table 1, Table 2, Table 3 and Table 4).

### 2.2. Data Collection

In October and November 2019, 23 pasture-based dairy farms (seven in Manawatu and 16 in Waikato) selected based on convenience sampling were visited by the first author. Of the 23 farms, 10 farms had a rotary milking platform and 13 had milked cows in a herringbone parlour. The number of milking cows in these farms ranged from 166 to 900 (see Appendix A, Appendix A) on the assessment day. On all farms, cows were kept at the pasture throughout the year using rotational grazing, with the length of rotation being dependent on the rate of grass growth. Of the 23 farms, 20 milked cows twice a day. On the remaining three farms (all in the Manawatu) cows were milked once a day.

On the twice-a-day farms, the process started ~2 h before afternoon milking on 17/20 farms. The first author undertook a questionnaire-guided interview which included questions related to record and management-based measures (see Appendix A) with a key worker (usually farm owner/manager), followed by the assessment of resources (Table 2) and then (Waikato herds only) behavioural observation of the milking herd at grass for ~0.5 h and, finally, recording of animal-based and stockpersonship measures (Table 1 and Table 2) during milking. On these 17 farms, a second assessor arrived at milking to undertake locomotion scoring. The first author was inside the milking parlour for the whole of the milking, whereas the second assessor stayed outside the parlour. On the remaining three twice-a-day farms, a second assessor was not available, so the assessment was carried out by the first author alone, with measurements at morning (locomotion scoring) and afternoon milking (remaining assessments).

On once-a-day farms, the assessment process started with the assessment of animal-based measures alongside locomotion scoring followed by the assessment of resources and, finally, the questionnaire-guided interview.

Alongside data collection, we assessed whether our proposed protocol was suitable for collecting data on all 32 welfare measures, during the time available (from ~2 h before milking to the end of milking). Additionally, we assessed whether our measures provided useful data that were relevant to the assessment of animal welfare on a farm and allowed comparison between farms.

### 2.3. Data Analysis

Data were analysed using SPSS version 27(IBM, Seattle, WA, USA). Descriptive statistics (mean, maximum value, minimum value, and 25th, 50th, and 75th percentiles) were calculated for all numerical animal-based measures (except where <0.5% of cattle were affected) and compared with their respective author-determined acceptable thresholds. The welfare category for each of those animal-based measures was then determined (Table 1) and a three-coloured heatmap was created. This comparison with author-determined thresholds and the creation of a heat map was then repeated for the categorical animal-based, resources-based, and stockpersonship-related measures.

## 3. Results

During the collection process, problems were identified with the collection of 3/32 measures included in the original protocol: maximum waiting time, distance to water points, and agonistic/positive behaviour assessment. Table 5 summarises the problems identified with each of these three measures and the changes needed for inclusion in the protocol. The only one of the three measures where data were collected on all farms where the collection was attempted were the counts of agonistic and positive behaviours. The results (raw counts) ranged from no behaviours observed (4/16 farms) to 26 behaviours observed. As these numbers were so low, no further analysis (e.g., calculation of behaviours observed/cow hour) was undertaken.

### 3.1. Animal-Based Measures

Descriptive statistics for the seven different animal-based parameters (with prevalence >0.5%, i.e., excluding blind eye and ingrown horn) are shown in Table 6, with a comparison to the author-derived acceptable thresholds (Figure 1). Overall mean percentage was above our acceptable range for broken tails, very dirty cows, and coughing during milking (observed percentages of 10, 17.3, and 1.2%, respectively, vs. acceptable thresholds of 5, 10, and 1%, respectively).

### 3.2. Resource-Based Measures

The heatmap for these measures is presented in Figure 2. Of the 23 farms, 16 had more than our acceptable threshold of 1.2 m^2^/cow in the collecting yard (range: 1.25–2.12 m^2^). For backing gate speed, in the farms with a circular yard (20/23 farms), eight farms had gate speeds above our ≤1 m/5 s threshold (range: 1.05–1.4 m/5 s), seven farms had gate speeds of exactly 1 m/5 s, and five farms had gate speeds below 1 m/5 s (range: 0.46–0.98 m/5 s). In all three farms with a rectangular collecting yard, the backing gate speeds were above our ≤0.5 m/5 s threshold (range: 0.96–1.51 m/5 s) (see Appendix A). Only (4/23) farms met our acceptable target for water trough cleanliness.

### 3.3. Track Assessment

On the 23 farms, a total of 48 tracks which ended at the collecting yard were identified and assessed for track width and surface. Of those 48 tracks, five tracks (on 5 farms) were recorded as having a poor surface. Only 45 tracks were assessed for camber (transverse slope) (three could not be assessed due to very poor track condition). Of these, 28 had a camber >8%. The minimum recommended track width varies by herd size (See: Efficient tracks. https://www.dairynz.co.nz/milking/milking-efficiently/cow-flow/track-and-yard/efficient-tracks/ (accessed on 1 December 2021)). Table 7 summarises the measurement of track width, camber, and track surface based on the largest herd size.

### 3.4. Shelter and Shade

Of the 23 farms, only two had shelterbelts which met the criteria for an effective shelterbelt (i.e., height >15 m and length at least twelve times tree height). Both had one East–West and one North–South shelterbelt. Only three farms had >80% of paddocks (enclosed grazing areas) with trees. The proportions of the largest grazing group which could be protected in these farms were 12.7%, 32.5%, and 39.7%.

### 3.5. Record, Management, and Stockpersonship Based Measures

The welfare levels for each of these measures are presented as stacked bar charts (Figure 3 and Figure 4). For stockpersonship-related measures, although 14/23 farms used some sort of handling aids to control their cows, handling of the cows on track by the stockperson was acceptable on all but two farms, where tail pulling and hitting were observed during milking.

## 4. Discussion

This study was an extension of a previous pilot study which developed a time-limited protocol for pasture-based dairy cows [6]. This study was designed to test the practicality and usefulness of that protocol on more farms across New Zealand. Three assessment measures (i.e., maximum waiting time, distance between waterpoints, and behavioural measures) were identified as having problems of feasibility or applicability. For maximum waiting time, this was because the in-parlour assessor had not been able to identify the time of arrival for all herds at the collecting yard on many of the farms assessed in the present study, although this was not the case in the pilot study [6]. This issue could be simply solved by ensuring that the outside assessor recorded herd arrival time and milking finish time alongside locomotion scoring. That assessor can record when a new group of cows enters the collecting yard and when the last cow of that group has finished milking. For the latter, the last cow in a group can be identified by the outside observer, as the farm staff will have to change gates to ensure that the new group (if there is one) goes to a different paddock than the previous one.

The remaining two assessments (distance between waterpoints and behavioural measures) require significant modification before inclusion. Distance between waterpoints was used by Kaurivi et al. [20] as a simple proxy for water availability. However, it was not applicable on these farms, as the rotational grazing system meant that cows were grazed on relatively small fenced-off paddocks [21]. In this study, paddocks generally only had one water trough, so the distance between water troughs could not be measured. Even though the paddock had two troughs, they were partitioned by fencing for grass management, so only a single trough was available for drinking. However, water availability is a key welfare issue, so alternatives are required. One potential alternative is to estimate the water requirements of the largest herd on a farm and then measure whether the trough can supply water at a sufficient rate to meet those requirements. Measurement of the refilling rate can be time-consuming, especially if multiple troughs are assessed. Another alternative is to measure trough capacity and ensure trough volume is at least half of the hourly demand (See: Farm Water Quantity and Quality. https://www.dairynz.co.nz/media/254175/5-15_Farm_water_quantit-and_quality.pdf (accessed on 1 December 2021)). Water requirements are very variable on New Zealand dairy farms [22], but an estimate of 70 L/cow/day supplied over 5 h (i.e., 14 L/cow/h) is likely to cover most situations. A quicker alternative, based on a suggestion by [22], is that at least 10% of the cows in a herd should be able to drink at once—equivalent to 50 cm of available trough space per cow. However, as identified by Jensen et al. [23], further research on the effect of these recommendations on competition, drinking behaviour, and water intake in dairy cows at pasture is required to establish that these are truly optimal recommendations.

Assessment of agonistic and positive behaviours also requires significant modification. Thirty minutes were set aside to observe behaviour while the cows were in the paddock before milking. No behaviours were observed on 4/16 farms and the total number of behaviours recorded was low. Additionally, the maximum rate of observed agonistic behaviour was only ~0.1 interaction/cow/h (assuming ~300 cows in each group, which is ~1/10 of the rate reported in cows at pasture by [24]). However, this rate calculation ignores the issue of identifying the true denominator when observing a group of 300 cows at pasture. One observer cannot simultaneously observe 300 cows, and it is likely that their attention will be principally drawn to cows that are nearer to the observer (i.e., close to the paddock exit). So the true denominator is likely to be less than 300 * 0.5 cow hours, but it is likely to vary depending on farm, paddock, and cow behaviour. One potential alternative is to simply record the number of cows that are easily observed and use that number as the denominator. However, it is likely that position within a field prior to milking is related to cow status [25], so simply recording behaviours of those cows may not reflect behaviour across the herd.

The low numbers observed may also in part be an issue of observation timing; [26] reported agonistic and grooming behaviour peaked when cattle returned to pasture after milking. However, even if the timing is changed, identifying differences between farms when the expected rate of behaviours is low [24,26] is likely to require a much longer observation period than used in this study. This would significantly add to the time required without necessarily adding value. The focus of the assessment of behaviours is to provide “indices of animals’ perceptions of their external circumstances” [27]. As such, it is unclear how useful assessment of behaviour at pasture when grass growth is at or near its peak (October/November in New Zealand [21]) is as an assessment of on-farm welfare. Measurement of behaviours when pasture availability is more likely to be restricted (e.g., in winter, which coincides with the late dry period and early lactation) is likely to be more useful, as feed restriction may be associated with increased aggression [28]. Such an assessment would thus need a separate visit in addition to the visit in October/November. However, even in such a situation measurement of the behaviour of cows at pasture for a short period just prior to milking is not a useful assessment. Further research is required on how best to assess behaviour-related welfare in cows at pasture, in particular, to determine what indicators should be used to identify positive behaviours [29], as well as to determine the optimal timing of such observations (during the day and across the year), the required observation duration, and the best way to observe a representative proportion of the herd.

This protocol was designed to be a time-limited assessment of milking cattle, so it is not aimed to be a comprehensive welfare assessment. It is thus inevitable that there will be compromises. Some of these compromises involve reducing the time spent assessing a particular welfare measure. This is particularly clear in regard to behavioural assessment, but it also applies to two of the animal-based assessments included in the final protocol, visual assessment of injuries and broken tails.

Visual assessments of injuries were assessed at a combined level, i.e., no separate scoring of cuts, abrasions, or swellings, or recording of the site of injury. This approach, while simple, lacks potentially useful details, and does not distinguish between cows with one or multiple injuries. Similarly, rather than using palpation, broken tails were assessed using visual assessment. This is faster but may be less sensitive and specific than palpation. If visual assessment of broken tails is to be used, its specificity and sensitivity compared to palpation need to be estimated.

Another issue is that this protocol is based on a single assessment point. In contrast to non-seasonal herds where cattle are at different points in their lactation cycle at the same point in time, in seasonally calving herds cattle are all at the same lactation stage. Thus, for measures such as body condition score and lameness, which vary considerably with the lactation stage [30,31], measurement at a single time point may not capture a farm’s true welfare status. Additionally, there are specific welfare challenges related to the season on New Zealand farms, such as cows self-feeding on root or leaf forages (e.g., fodder beet) during winter (See: End Intensive Winter Grazing. https://www.youtube.com/watch?v=9sJZgQ3yNTA (accessed on 1 December 2021)) or pasture restriction, which will be missed if the assessment is restricted to just one timepoint within a year.

Nevertheless, we believe that, given these constraints, this is a practicable and feasible protocol that can be used to provide a reasonable assessment of the welfare of pasture-based cows in a single visit. However, further testing on more farms with more assessors in order to provide details on repeatability and reliability of the measures included in our protocol are required.

The thresholds used in this study were based on the author’s opinion and are not intended to be definitive, but rather a starting point for discussion. Additionally, this was a convenience sample of farms in two regions in New Zealand, so we do not know how representative the results from our study farms are of farms across New Zealand. Nevertheless, using our thresholds, it was clear that shelter and shade were issues on all the dairy farms examined, with none of the farms having acceptable areas of either shade or shelter. In a pasture-based system, it is recognised that shade and shelter are important for at least part of the year [32], but they seem to be a low priority on dairy farms in New Zealand. Comparing these results to those from [15] on beef farms on the North Island of New Zealand, it is instructive that the only farm that they recorded as having insufficient shelter was a farm that was being converted to beef from dairy.

The other resource-based assessments in which a high proportion of study farms were identified as being in either our marginal or unacceptable category were the cleanliness of drinking troughs and track condition. For the former, 11 out of 23 farms had at least one unacceptably dirty trough. This was principally because most farms which used a feed pad (a designated, usually concrete, area, for feeding supplementary feeds, e.g., concentrates, palm kernel extract, and/or silages) had dirty troughs, as animals eat the supplementary feed and then drink from and contaminate the troughs. Our data thus suggest that more attention needs to be paid to managing this issue. Track quality was marginal or worse on 21/23 farms. As a major factor associated with the development of lameness [33], it is disappointing that track quality was not better. However, it is possible that the high proportion of farms categorised as marginal or unacceptable reflects a very rigorous assessment of track conditions. Despite the study being undertaken at the expected time of peak lameness prevalence in North Island dairy herds [30], all of the herds had <10% lameness with 20/23 having <5%. This reflects how animal-based measures indicate actual welfare problems and resource-based measures indicate potential welfare problems.

For the animal-based assessments, comparisons of percentiles and the author-imposed thresholds identified three conditions where mean and median percentages were outside our acceptable ranges (Table 6). These were dirtiness, coughing, and broken tails. Dirtiness was also identified by Fabian et al. [30] as an issue on the pasture-based dairy farms that they assessed in New Zealand. However, as those authors stated, in contrast to housed cows where dirtiness often reflects faecal contamination, under New Zealand’s conditions dirtiness reflects soil contamination, which probably has less of a welfare impact than faecal contamination. Nevertheless, especially with the concern around winter grazing and mud (See: End Intensive Winter Grazing. https://www.youtube.com/watch?v=9sJZgQ3yNTA (accessed on 1 December 2021)), we need more data on the impact of cleanliness of New Zealand cows on other welfare outcomes (such as mastitis). For coughing, further research, particularly on the cause of the coughing, is required to better understand its welfare impact. This research could then be used to provide data for a more objective welfare threshold for coughing in pasture-based dairy cattle.

The final condition identified as having a median within-herd prevalence higher than the upper threshold of the acceptable welfare zone was broken tails. Despite tails being observed rather than palpated, which would miss non-deviated broken tails with small swellings, we still observed broken tails in ~10% of cows. These results are consistent with previous reports from New Zealand [14] and support their conclusion that more needs to be done to reduce the prevalence of broken tails in New Zealand dairy cattle.

This is a descriptive study undertaken on a convenience sample of farms. Thus, we need more data from a representative, preferably randomly sampled, sample of farms across New Zealand to better establish how our thresholds are reflected on dairy farms across New Zealand. Such a study could be undertaken alongside the assessment of the repeatability and reliability of the measures included in our protocol.

## 5. Conclusions

This study has shown that the time-limited protocol we developed in a previous study was generally feasible to use across a range of farms. We believe that after further testing for repeatability and reliability with multiple observers and further discussion and testing of thresholds for determining whether welfare is acceptable or not, this protocol can form the basis of a welfare assessment protocol for pasture-based dairy cattle in New Zealand and elsewhere.

## Figures and Tables

**Figure 1 animals-12-02481-f001:**
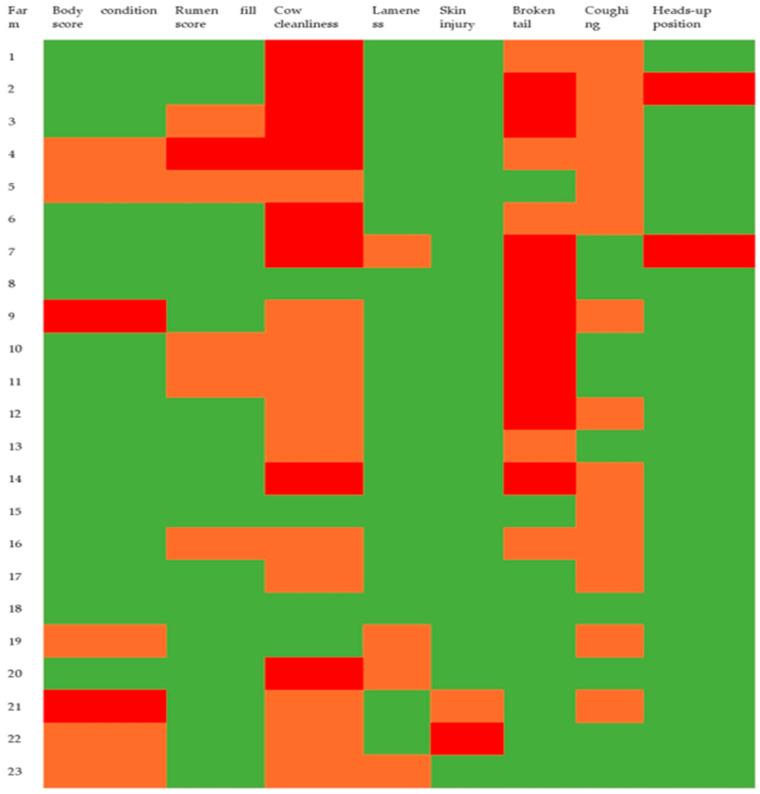
Heatmap representing welfare levels for eight animal-based measures from 23 dairy farms. All measures have three welfare levels (acceptable, green; marginal, orange; or unacceptable, red), except for the heads-up position which has only two levels (acceptable, green; and unacceptable, red). See Table 1 for further details of measurements and thresholds.

**Figure 2 animals-12-02481-f002:**
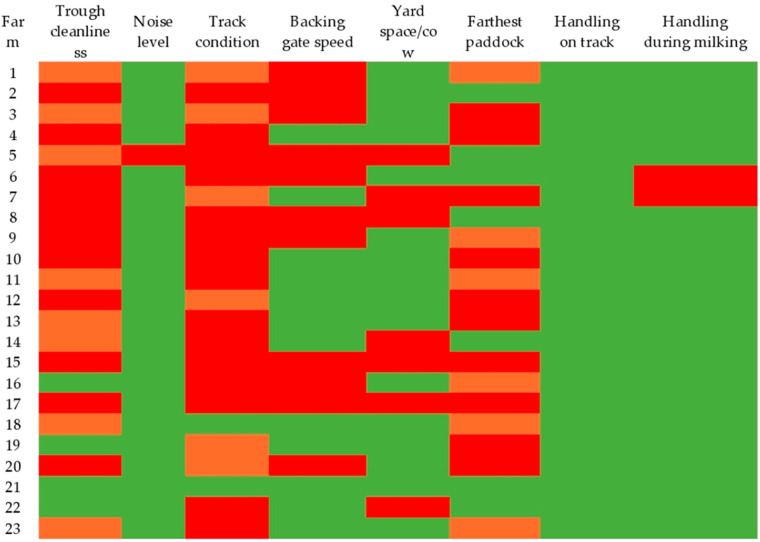
Heatmap representing welfare levels for six resource-based measures and two handling measures. All measures had three welfare levels (acceptable, green; marginal, orange; or unacceptable, red), except for yard space, backing gate speed, and the two handling measures which had two levels (acceptable, green; unacceptable, red). See Table 2 for further details of measurements and thresholds.

**Figure 3 animals-12-02481-f003:**
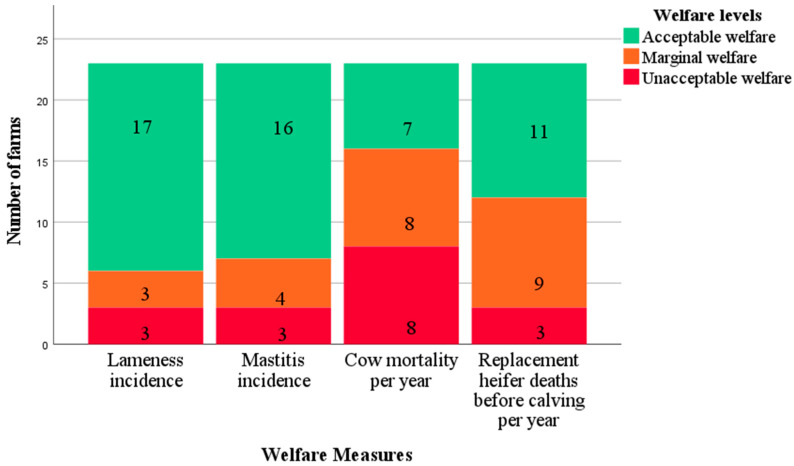
Stacked bar chart for welfare levels of disease and mortality data for 23 pasture-based dairy farms.

**Figure 4 animals-12-02481-f004:**
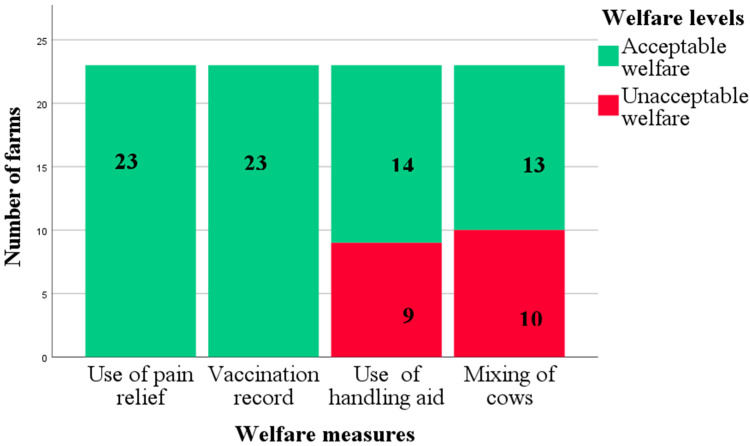
Stacked bar chart for welfare levels of management data for 23 pasture-based dairy farms.

**Table 1 animals-12-02481-t001:** Method of assessment for animal-based welfare measures from the pasture-based dairy cattle welfare assessment protocol, with categorisation based on author-determined thresholds. Note: all cows assessed for all measures, and no sampling used.

Measure	Basis of Scoring	Animal Welfare Categories		
		Acceptable	Marginal	Unacceptable
Body condition Percentage of cows ≤3.5.	1–10 score (See: Body Condition Scoring. www.dairynz.co.nz/animal/body-condition-scoring/how-to-bcs/ (accessed on 1 December 2021)).	Green: 0–5%	Orange: >5–10%	Red: ≥10%
Rumen Fill Percentage of cows ≤2.	1–5 score (See: Rumen Fill Scorecard. https://dairyveterinaryconsultancy.co.uk/download/rumen-fill-scorecard/ (accessed on 1 December 2021)).	Green: 0–2%	Orange: >2–5%	Red: >5%
Cleanliness Percentage of very dirty cows.	Three-category cleanliness score: clean, dirty, and very dirty. Scored as^,^ with modifications: only flank and hind quarters were observed with cow scored based on worst score [17].	Green: <10%	Orange: 10–20%	Red: >20%
Heads-up position	Assessment of cows standing in the collecting yard with their heads up ^a^ immediately after one movement of the backing gate.	Green: No heads up		Red: Any heads up
Lameness Percentage of cows score ≥2	0–3 score (See: Lameness Scoring. https://www.dairynz.co.nz/animal/cow-health/lameness/lameness-scoring/ (accessed on 1 December 2021)).	Green: 0–5%	Orange: >5–10%	Red: >10%
Broken Tail Percentage of cows.	Deviation/swelling of the tail visually assessed from within 2 m of the cow during milking. Any visible deviation or swelling in any region of the tail was recorded as a “broken tail”.	Green: 0–5%	Orange: >5–10%	Red: >10%
Coughing Percentage of cows.	Cows coughing while being milked.	Green: 0–1%	Orange: >1–2%	Red: >2%
Skin Injury Percentage of cows.	Visual assessment of abrasions, cuts, hairless patches, and swellings was assessed at a combined level, i.e., no separate scoring of abrasions, cuts, hairless patches, and swellings. Observed from behind in rotary and at exit in herringbone. Any visible injury was recorded as injured.	Green: 0–1%	Orange: >1–2%	Red: >2%
Ingrown horn Percentage of cows.	All cows with overgrown horns which could penetrate the skin in future were recorded, along with horns that had already penetrated the skin.	Green: 0–1%	Orange: >1–2%	Red: >2%
Blind eye Percentage of cows.	All cows that were blind in one eye or had visible eye damage were recorded.	Green: 0–1%	Orange: >1–2%	Red: >2%
Agonistic Behaviour Total number of agonistic behaviours observed.	A 30 min observation of all cows in the paddock (current grazing area). All social agonistic behaviours (head butting, chasing, displacement, fighting, pushing) were recorded (>1 s elapsed—new behaviour).	ND		
Positive Behaviour Total number of affiliative behaviours observed	A 30 min observation of all cows in paddock. All social affiliative behaviours (allogrooming, lapping) were recorded (>10 s elapsed—new behaviour).	ND		

^a^ cows normally stand with their heads at the same level as shoulders, “heads up” indicates a cow is being pushed and space is limited [18]; ND, not determined.

**Table 2 animals-12-02481-t002:** Method of assessment for resources and stockperson-based measures from the pasture-based dairy cattle welfare assessment protocol, with categorisation based on author-determined thresholds.

	Basis of Scoring	Animal Welfare Categories		
		Acceptable	Marginal	Unacceptable
Distance to waterpoints	Distance between water points/troughs in the grazing area.	Green: ≤150 m	Orange: >150–250 m	Red: > 250 m
Shade availability Proportion (%) of cows with shade available if grazed by largest herd.	If < 80% of paddocks planted with trees, welfare was recorded as unacceptable. If >80%, paddock next to one being currently grazed was assessed for available shade at noon [19] (see appendix 3 for details). Required shaded area was set at of 6 m^2^/cow (See: Trees for Shade. Retrieved from https://www.dairynz.co.nz/media/5447835/Trees_for_shade.pdf (accessed on 1 December 2021))	Green: >40%	Orange: 20–40%	Red: <20%.
Shelter availability (against wind) Proportion of cows with shade available if grazed by largest herd.	Shelterbelts (trees at least 15 m high and length of the belt at least 12 times the height of the trees) were recorded.	Green: Shelter for 100% cows	Orange: Shelter for at least 50% of cows	Red: Shelter for <50% cows.
Maximum waiting time in collecting yard	Time of arrival of each milking group and the time of exit of the last cow of each group from the milking parlour were recorded.	Green: ≤2 h	Orange: 2–2.5 h	Red ≥2.5 h
Trough cleanliness	Visual observation of where cows are grazing and one adjacent paddock. Troughs at the feed pad were assessed when present. The worst observation was recorded.	Green: Clear water, easily visible base, no apparent dirt/dead insects.	Orange: Base partly obscured, no floating dirt or dead insects.	Red: Base not visible, dirt/insects
Noise level	Subjective assessment of noise inside the milking parlour.	Green: Minimal noise, conversation can be easily herd	Orange: Some noise, conversation heard with concentration.	Red: High noise, conversation cannot be heard.
Furthest paddock distance	Farmer-estimated distance from the milking parlour to the furthest paddock.	Green: <1 km;	Orange: 1–1.5 km	Red: >1.5 km
Track condition	Assessment of width, camber, and surface quality of the track/s (See: Efficient tracks. https://www.dairynz.co.nz/milking/milking-efficiently/cow-flow/track-and-yard/efficient-tracks/ (Accessed 1 December 2021)), (See: DairyNZ Improving Cow Flow. https://www.youtube.com/watch?v=XUnzfH9JpPM&ab_channel=DairyNZ/ (Accessed 1 December 2021)) leading to the parlour (100 m). If multiple tracks, the worst case was used for categorisation.	Green: All three track assessments good	Orange: Two track assessments good, must include track surface	Red: Two track measures poor/poor track surface
Yard space per cow	Available space/cow in the collecting yard. Note: was checked against the largest herd size of the farm (See: Yard Features. https://www.dairynz.co.nz/milking/milking-efficiently/cow-flow/track-and-yard/yard-and-handling-facilities/yard-features/ (Accessed 1 December 2021)).	Green: >1.2 m^2^/cow;		Red: ≤1. 2 m^2^/cow
Backing gate speed	Speed of the backing gate (See: Backing gates. https://www.dairynz.co.nz/milking/milking-efficiently/cow-flow/track-and-yard/yard-and-handling-facilities/backing-gates/ (Accessed 1 December 2021)).	Green: ≤1 m/5 s for circular yard and ≤0.5 m/5 s for rectangular yard		Red: Gate speed greater than those limits
Handling on track	Observation from behind stockperson as cows were brought in for milking. Pressure identified when cows lifted their heads or ran in response to stockperson/dog.	Green: No pressure observed		Red: Pressure observed

**Table 3 animals-12-02481-t003:** Method of assessment for record-based assessment measures from the pasture-based dairy cattle welfare assessment protocol, with the categorization based on author-determined thresholds.

	Basis of Scoring	Animal Welfare Categories		
		Acceptable	Marginal	Unacceptable
Lameness incidence	Percentage of milking herd treated for lameness per year.	Green: 0–10%	Orange: >10–15%	Red: >15%
Mastitis incidence	Percentage of milking herd treated for mastitis per year.	Green: 0–10%	Orange: >10–15%	Red: >15%
Cow mortality	Percentage of milking herd that die on farm in a year.	Green: 0–1%	Orange: >1–2%	Red: >2%
Replacement heifer deaths before calving/year	Percentage of replacement heifers that die before calving for the first time.	Green: 0–0.5%	Orange: >0.5–1%	Red: >1
Vaccination record	Presence or absence of vaccination records.	Green: present		Red: absent
Pain relief	Use of pain relief before routine husbandry practices (disbudding, dehorning, castration, removal of supernumerary teats).	Green: use of pain relief in all calves for all procedures		Red: Pain relief not used for all calves/procedures

**Table 4 animals-12-02481-t004:** Method of assessment for stockpersonship measures obtained from questionnaire, with the categorization based on author-determined threshold.

	Basis of Scoring	Author’s Welfare Categorisation		
		Acceptable	Marginal	Unacceptable
Mixing of Cows	Mixing of cows between herds if more than one herd on a farm or mixing of cows that are new to the farm (i.e., purchased cows). Does not include mixing of heifers after calving with lactating cows.	Green: No mixing of cows	Orange: Rare mixing of cows (up to two times per lactation)	Red: Frequent mixing of cows
Use of handling aids	Use of handling aids (during milking) such as prod, wooden/rubber pipe Note: if use is observed during the assessment, it will be prioritised over the farmer’s questionnaire response.	Green: No use of mentioned handling aids		Red: Use of handling aids

**Table 5 animals-12-02481-t005:** Welfare measures found not to be feasible on all farms and reasons why they were not.

Measures	Issue Identified	Solution
Distance to waterpoints	Most paddocks only had a single trough, and on some farms, even though the paddocks had multiple troughs, the cows only had access to one trough because of temporary electric fencing.	Alternative method of measure of water availability is required. Further research is required to establish a valid but not time-consuming assessment method
Maximum waiting time in the parlour	Main assessor inside the parlour could not record the arrival of multiple herds if present.	The outside assessor can record the arrival of a new group at the collecting yard and when the last cow of a group leaves the milking parlour alongside locomotion scoring
Agonistic/positive behaviour	The 30 min allocated to this observation was insufficient, with no observations of agonistic or positive behaviour on many farms. Calculating the denominator (i.e., the number of cows being observed) was also difficult as it was not possible to observe all cows in a paddock simultaneously.	Further research is required to establish a more effective and valid method of identifying the behaviour of cows at pasture. Simply increasing observation time would be unlikely to provide useful data and would conflict with the aim of producing a time-limited protocol.

**Table 6 animals-12-02481-t006:** Descriptive statistics for selected animal-based measures from 23 farms and their comparison with a predetermined acceptable threshold (see Table 1 for details of measurements).

Measure	Mean (%)	Max (%)	Min (%)	Percentiles			Acceptable (Green) Threshold
				**25**	**50**	**75**	
BCS (≤3.5)	4.8	14.2	1.2	2.5	4.1	**5.7**	≤5%
RFS (≤2)	1.9	7.8	0.0	0.0	1.2	**3.4**	≤2%
Lameness (≥2)	3.3	9.3	0.4	1.2	2.8	4.6	≤5%
Broken tails	**10.0**	24.3	0.3	2.9	**9.2**	**17.8**	≤5%
Dirtiness	**17.3**	38.8	2.7	**11.6**	**15.7**	**23**	<10%
Skin Injury	0.4	2.4	0.0	0.1	0.3	0.6	≤1%
Coughing	**1.2**	2.2	0.0	0.8	**1.3**	**1.4**	≤1%

BCS, body condition score; RFS, rumen fill score. Figures in bold where mean or percentiles exceed the upper green threshold.

**Table 7 animals-12-02481-t007:** Measurement of width, camber, and surface of the tracks leading to the parlour based on the largest herd size of the farm.

Farm No	Group of Herd Size	Largest Herd Size	Recommended Width (m)	Track(s) Width (m)	Camber (cm/m)	Bad Surface (in Number)
5		224		5.4, 4.2	4.6, 4.3	1
6		224		4.5	8.7	
8		210		5.5, 4.4	11, 8.5	
9		240		4.8, 4.1, 6.6	20.2, 22.5	1
12		178		5.8, 6	9.8, 11	
14	(120–250)	235	5.5	5.2, 5.3, 5.1	20.8, 17.1	1
18		240		6, 5.5	6.8, 6.6	
19		125		5.2, 5.5	7.6, 21	
21		216		8.2, 5.9	7.8, 6.2	
22		166		3, 4	8.2, 12	
1		305		4.6	6.7	
3		300		4.5, 3.9	5.7, 4	
4		257		4.5, 4.1	8.1, 8.6	
11	(251–350)	310	6.0	5.4	8.6	
16		350		7.8, 5	8.8, 5	
17		350		4.6, 5.5	7.8, 14, 1	
20		330		6.9, 6, 6	9.2, 5.2	
23		364		3, 4.3, 4.5	9.4, 8.4, 7.8	
13		350		5.2, 5.3, 5.1	8.6, 8.5, 7.2	
15		364		5.6, 4.6	9.8, 6.7	
10	(351–450)	360	6.5	6, 4.1, 4.5	12, 9.4, 8.4	1
7		450		8.2	10.25	
2		420		4.7, 5	9.5, 6.7	

## Data Availability

Not applicable.

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
