# Peer review of "Practicability of a Time-Limited Welfare Assessment Protocol for Pasture-Based Dairy Farms, and a Preliminary Assessment of Welfare Outcome Thresholds"

_animals, 2022, doi:10.3390/ani12182481_

Round 1

Reviewer 1 Report

I have added comments and suggestions with the attached PDF. Overall a practical work that deserves to be shared.

In that context, further practicality should be incorporated and considered.

Clarity is needed - is the intent of this assessment to be used as a market tool and to help direct farms where they need to work on improvements in order to maintain their access to a milk market, or as a purely scientific endeavor to precisely measure animal welfare ? If the former that I would argue that an argument could be made from the results presented that several measures be dropped (coughing, injury, RFS, gate speed...or make a more clear case for it ),  refined (cow cleanliness as it is not clear what you are trying to measure, the quality of the lying area or where they walk ) ,  or added (moderate lameness) and that a new sample approach should be considered as it will be largely impractical to assign 2 evaluators to a farm. 

 Work on interrater reliability would likely direct you to refine the scoring system to a simple 1-2-3 option for all outcomes (not broken tails) as you have for lameness. The current BCS method is intended for managing nutrition and is too complicated, for reliability it is best to avoid such a wide scale.

Additional background should be provided on the data collected from the farm and the reliability and accuracy of it.  Written or computer based ? We find that most of the herd data is not useful as between farm definitions are variable enough that it is difficult to report on across herds in a meaningful way. Other outcomes based on the questionnaire should also provide guidance on how the answer is verified, if it is.

Deeper discussion on the practicality of measuring tree shade if one expects users to be willing to apply it.  And how does one consider that the calculation will be dependent on the time of day.

Reviewer 2 Report

Sorry for the technical comment, but is there a dot at the end of the title? I admit that I have not come across such a solution.

I am in favor of this approach to also write in the Abstract what was the purpose of the research / study. I found the formulated goals of the study in the last paragraph of the Introduction chapter. However, in this case, I am careful to write down what was the cognitive (scientific) goal and what was the utilitarian (useful) goal when formulating the research goal (s). A clearly defined scientific and useful goal will allow to state in the Conclusions chapter whether these goals were achieved as a result of the research / study. On this basis, it is possible to indicate the scope of further research to be carried out in the future.

In the final part of the Introduction, it would be worth formulating a research problem based on the review of the state of knowledge. You can also indicate a gap in the current state of knowledge. Thanks to this, the Authors can present excellent premises for the formulation of the purpose (s) of the research study.

If research is related to animal welfare assessment on dairy farms, it would be helpful to write a paragraph or two about welfare in the Introduction. The theory and practice on this subject is very rich in terms of scientific publications. Therefore, it is worth developing the issues of the importance of animal welfare, in this case dairy cows, which justify the authors of the proposed research.

In the Materials and Methods section, I failed to provide more details regarding the description of dairy herds in the farms visited. It would be worth mentioning the milk yield of cows, how long (how many months in a year) the cows stay on the pasture, how the passage of cows between quarters on the pasture is organized, how much time the cows stay in one quarters on the pasture. In addition, it would be worth providing details of models of cow milking equipment used on farms, taking into account selected construction data, the number of milking stalls, and herd management systems.

I would like to know on what basis the authors adopted the percentage ranges for the considered variables and the Green, Orange and Red options? (Tables 1, 2 and 3).

In Table 1, in the first line, instead of Redc: ≥ 10%, there should be Redc: > 10%. The same remark applies to the first line of Table 2; instead of Redc: ≥ 250m there should be Redc: > 250m. There are similar problems in the other lines of Table 2. 

In line 144 the authors wrote: Data were analyzed using SPSS v27. It would be good to explain what is meant by SPSS v27. It is certainly the name of a statistical program, but to be on the safe side, it would be worth giving the reader more details. If it is a statistical program, you must provide citation / affiliation details, i.e. company, city, country (in References).

I think it would be a good idea to make comparisons of the results obtained with the standards / recommendations for some parameters of the production environment in the dairy farm. The authors present details of the data on the milking installations in the farms visited. These parameters can be assessed on the basis of a comparison with standards and recommendations, for example citing the study: Do International Commission of Agricultural and Biosystems Engineering (CIGR) dimension recommendations for loose housing of cows improve animal welfare? The topic of comparisons in different dairy cattle production zones is also developed in the publication: Method for comparing current versus recommended housing conditions in dairy cattle production. In my opinion, by developing the discussion of the research results (in the Discussion chapter), one can point to recommendations that are important for the improvement of dairy production conditions in the researched farms.

In my opinion, the Conclusions could be a bit more extensive and refer to the results of the research study carried out.

In References, the articles cited should include the full list of authors, not the name of the first author and the abbreviation et al.

In the Supplementary materials, the Authors cited publications that were not included in the References, e.g. [Armson, 2012]. These deficiencies need to be supplemented. A similar situation is in the text of the article; The authors cite on page 7: Monteith and Unsworth (1990), but this publication is not in References; this also needs to be completed.

In the text of the article, the Authors refer to Figures, and even take into account captions to Figures, but I did not have the opportunity to see Figures and assess their value. I just didn't have access to Figures.

I would like to ask whether, in the context of the research carried out, farmers could independently assess the welfare of cattle on their own dairy farm? 

Round 2

Reviewer 2 Report

Thank you for including changes and additions to the text, in line with the suggestions in the review.